# GROUNDING CODE UNDERSTANDING IN STEP-BY-STEP EXECUTION

## ABSTRACT

Auto-regressive language models have made significant inroads in code generation, reasoning, and execution in recent years. Despite the recent progress, however, even the most capable models have been shown to perform significantly worse than humans in the task of predicting what a given piece of code does. This has fueled concerns about the tendency of models that seemingly generate and reason over code to learn shortcuts without developing any deeper understanding of code. Unlike reasoning, the meaning of a line of code is determined entirely by the effect it has on the state of the machine on which it is executed. Inspired by this observation, we propose measuring code understanding as the ability to predict the effects of line-by-line execution of a piece of code. We perform an empirical study which suggests that the inability to track machine state is a key contributor to the deficiencies of existing models to understand code. We also propose a simple solution based on fine-tuning a model on auxiliary state supervision, and we demonstrate the effectiveness of this approach.

## 1 INTRODUCTION

Classic computers execute code by sequentially processing a series of instructions, updating memory at each step. In contrast, when a Large Language Model (LLM) or a human is tasked with executing code, they might take shortcuts by skipping intermediate steps and directly predicting the outcome. While such shortcuts can in some cases reveal a deep understanding of the code, they can also be misleading and error-prone, in particular, when they rely on superficial aspects of the code, such as function or variable names , or merely relates the structure of the code to examples seen during training. This raises the question to what degree a model trained to execute or describe code can truly "understand" the code at the instruction level. If not, how can such an understanding be instilled or improved in a given model?

To illustrate this point further, we consider the following simple Python program:

```python
def factorial(n):
    if n == 0:
        return 1
    else:
        return n * factorial(n - 1)
```

We can find the results to the inputs 1 to 20 on the Wikipedia page on Factorial (Wikipedia contributors, 2024). Prevalent LLMs have seen this during training (Merity et al., 2016) and might correctly recite the results. But when we ask for the output to `n` above 20, this shortcut no longer works and results in an incorrect prediction. Actually executing the code guarantees the correctness of the result, but also requires that the model execute each of the steps in sequence, and keep track throughout of the value of `n`. If the model did rely on step-by-step execution and made a wrong prediction we would have no way of knowing where the mistake happened. The recursive function hides multiple program state transitions and makes it difficult to locate errors.

With these observations in mind, we present a code execution benchmark and study, centered on the idea of *step-by-step* execution of code without control flow elements. In other words, the code we consider only contains statements in each line, and as such, after each line the state of the program

Table 1: Examples of how we transform a function into code trace.

| Code in function | Traced lines | Code trace |
|---|---|---|
| ```for v in range(2):```
```    w = v + 1``` | ```for v in range(2):```

```w = v + 1```
```for v in range(2):```
```w = v + 1``` | ```forloop0 = iter(range(2))```
```v = next(forloop0)```
```w = v + 1```
```v = next(forloop0)```
```w = v + 1``` |
| ```if c:    # c == False```
```    b = 1```
```else:```
```    b = 0``` | ```if c:```
```else:```
```b = 0``` | ```b = 0``` |
| ```while c:    # c == True```
```    c = False``` | ```while c:```
```c = False```
```while c:``` | ```c = False``` |

and its namespace are unambiguous. We can construct such step-by-step computations by unrolling the execution of a function on some input, while tracing the code lines. This ensures that we retain the same expressivity, in terms of input-output distribution, that is found in existing code execution benchmarks.

For code execution, any given instruction manifests solely through the change it causes in the machine's state. Therefore, understanding and tracking these state changes is crucial for code execution. A model's ability to execute code step by step can be thought of as functionally equivalent to its ability to track state. State tracking is not only important for code execution but also extends to tasks like document comprehension and general agentic tasks. In this work, we focus on code execution as an expressive and verifiable test bed for state tracking. This allows us to isolate and study state tracking as a fundamental capability of LLMs.

Empirically, we find that large language models (LLMs) struggle with programs that consist of many steps and with programs containing individual steps that hide significant complexity in a single command. The latter (single-step complexity) relies on memorization, or further unrolling into less complex steps. This constitutes a dimension along which code execution is challenging for any given model. However, the first (execution length) is a dimension where significant improvement should be possible by appropriate training. Overall, this highlights that there exists a significant semantic gap in existing language models' understanding of code.

To bridge this gap, we propose using explicit state supervision to induce step-by-step state transitions in the hidden representations. Our results suggest that state supervision improves a model's ability to understand and track state throughout program execution, thereby enhancing its code execution performance. This highlights that step-by-step execution could be an important and largely overlooked ingredient to improving code understanding abilities in LLMs.

## 2 CODE TRACES FROM FUNCTIONS

We assess code execution in Python. Given some Python code and input values, we ask the LLM to predict the output. Following Gu et al. (2024) we also consider the reverse: given the output, predict a suitable input value. In both settings, the prompt to the LLM consists of the source code and either the input or output representations.

The amount of compute a decoder-only Transformer applies scales quadratically with the prompt and prediction length, which is not enough for many problems. For example, a naive solution to the Traveling Salesman Problem tries all possible permutations of cities resulting in a complexity of $\mathcal{O}(n!)$. Merely on the grounds of insufficient compute, we can already say that this is impossible to simulate for the Transformer. Comparing the code execution performance in such a setting might only measure the degree to which the LLM memorized examples instead of its step-by-step execution capability.

Table 2: Examples of code traces from each of our three benchmarks

| CRUXEval | MBPP | Arithmetic programs |
|---|---|---|
| ```text = '123'```
```text_arr = [],```
```forloop0 = iter((range(len(text))))```
```j = next(forloop0)```
```text_arr.append(text[j:])```
```j = next(forloop0)```
```text_arr.append(text[j:])```
```j = next(forloop0)```
```text_arr.append(text[j:])```
```assert (text_arr) == ['123', '23', '3']``` | ```arr, n = [1,2,3,1,1],5```
```arr.sort()```
```prod = 1```
```forloop0 = iter((range(0,n,1)))```
```i = next(forloop0)```
```i = next(forloop0)```
```i = next(forloop0)```
```prod = prod * arr[i]```
```i = next(forloop0)```
```prod = prod * arr[i]```
```assert (prod) == 6``` | ```d, a, c = 4, 3, 6```
```a = 9 - a```
```a = d - a```
```b = c - 4```
```a = b + a```
```assert a == 0``` |

We address this issue by replacing the function with its unrolled version: given the function and an input we trace each line in the function whenever it is executed. This turns the problem setting into one where the amount of compute applied by the Transformer grows proportionally with the amount of steps the function runs for.

We run the function on the input and log a line of code whenever it is executed. Then, we apply a set of transformations on the sequence of code lines. These transformations turn the sequence of code lines into a valid Python program, which we refer to as *code trace* (Zhang et al., 2024). When a line consists of an assignment, we copy it unaltered into the code trace. When a line contains control flow elements, we skip any expressions that do not alter the values of any variable. See Table 1 for a list of examples on how we transform code in functions to code traces.

Any dataset that contains both Python functions and example inputs can be automatically turned into a traced version; alternatively, one can also generate synthetic code traces. We perform our experiments on two code trace datasets derived from functions of the CRUXEval (Gu et al., 2024) and MBPP (Austin et al., 2021) datasets, as well as one set of synthetic traces, which we refer to as Arithmetic Programs. We refer to Table 2 for one trace example from each. See Appendix C for details on how we generate the Arithmetic Programs.

When it comes to benchmarking code execution, the first two datasets are in some sense at the opposite ends of a spectrum: CRUXEval was only recently introduced, its functions are synthetically generated, they do not carry an intuitive name in their definition, and are designed to lack a clear intent or task they should be performing; MBPP has been introduced in 2021, its functions are sourced from human programmers, and each of them is paired with a task description, which is hinted to in the respective function name; they therefore make for an ideal test-bed for investigating code execution shortcuts. Our Arithmetic Programs, conversely, are designed to enable systematic investigation of code execution capability at different levels of trace length and complexity.

## 3 EVALUATION ON CODE TRACES

In this section we evaluate the code execution performance on code traces on a suite of pre-trained LLMs, including both models specialized to code and generalist models.

**CRUXEval-I/O** We compare the performance of multiple LLMs on the traced and non-traced functions on the input and output prediction benchmarks of CRUXEval. For the non-traced case, we adopt the numbers from the leaderboard[1]; for evaluating on traces (and later on MBPP) we fork the CRUXEval code[2]. We filter out all samples whose traces that are longer than 50 steps. See Figure 1 for some useful statistics on the traced benchmark.

In Figure 2 the results show that the traced version generally achieves a higher accuracy compared to the baseline for both input and output prediction. In Figure 3 we observe that there is a high correlation between the traced and non-traced version, as expected: in order to figure out the output or input of a CRUXEval function, the model must, for the most part, actually execute its code; and it has an easier time doing so when this is unrolled into a code trace, as we speculated in section 2.

---

[1] https://crux-eval.github.io/leaderboard.html
[2] https://github.com/facebookresearch/cruxeval

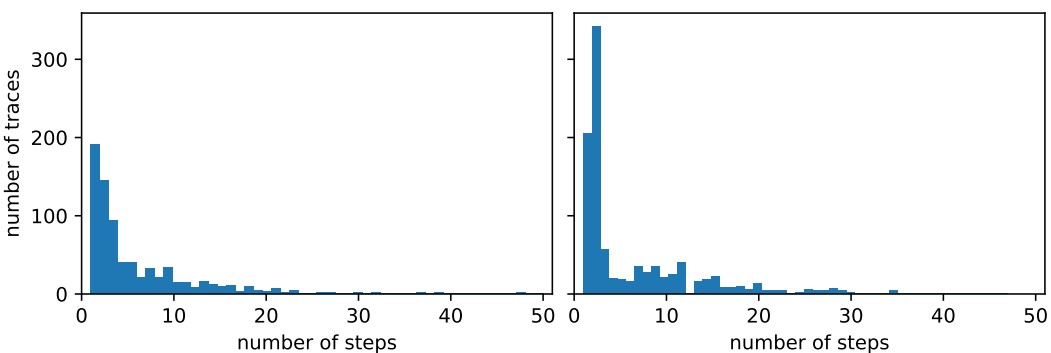

Figure 1: Distribution of trace lengths. CRUXEval (left): 761 execution traces from 761 programs. MBPP (right): 999 execution traces from 347 programs.

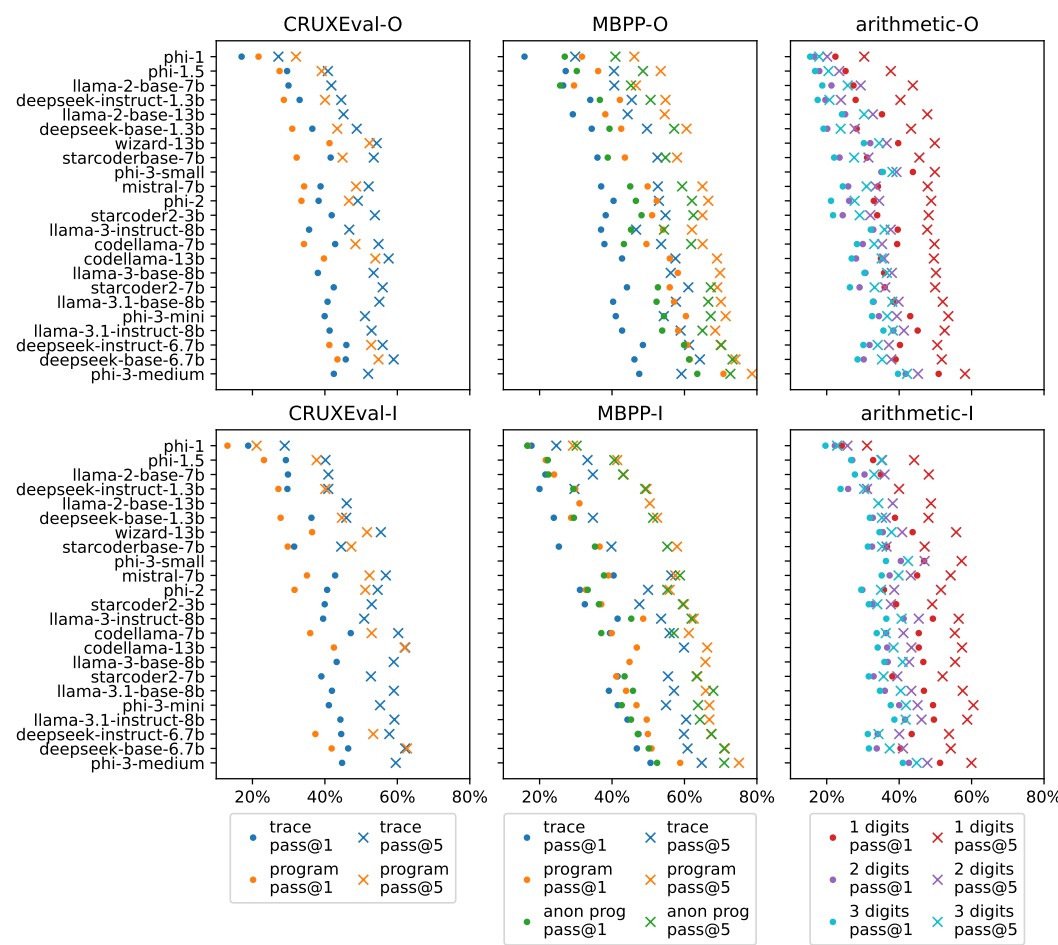

Figure 2: Evaluating LLMs at code execution. For CRUXEval, all models perform better on code execution and understanding when faced with the traced version of a function, rather than the function itself; the opposite is true for MBPP. For arithmetic programs, more digits make the problem harder for all models.

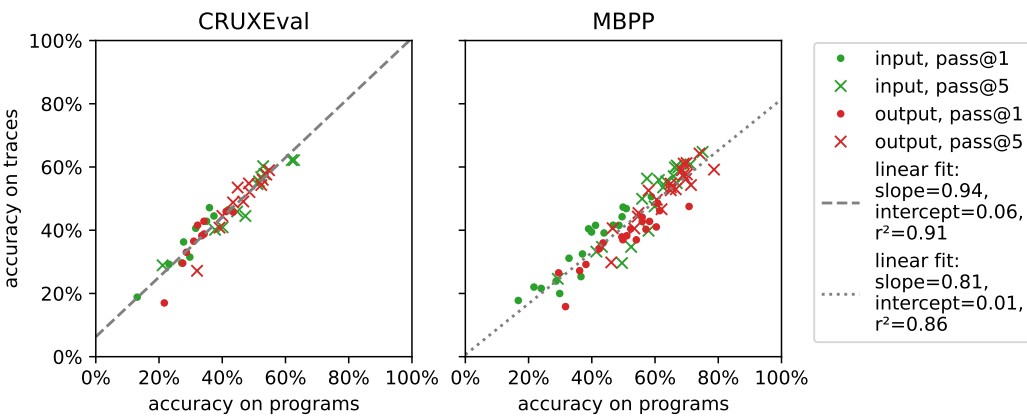

Figure 3: The correlation between accuracy in executing a program and executing its trace is strong for CRUXEval, less so for MBPP.

**MBPP-I/O**   We repeat the same analysis for the MBPP benchmark. In order to turn it into a code execution benchmark in the vein of CRUXeval, we take the three unit tests associated to each sample/function and grab from these the associated inputs and outputs; these are then used in the same way as in CRUXEval to run input and output prediction tasks, as well as to obtain the traced version of the benchmark. We report the details of this procedure in Appendix A, and again statistics on the traced version in Figure 1.

We report in Figure 2 our results, which paint a different picture than the ones gathered on CRUXEval: the non-traced version of the benchmark generally achieves higher accuracy compared to the traced one, on both input and output prediction; moreover, one can observe (see Figure 3) a weaker correlation between performance on the direct benchmark and performance on its traced version. We can also observe the performance on the non-traced benchmark is higher overall than on CRUXEval, with scores up to 80% on output prediction. MBPP is an objectively *easier* code execution benchmark.

**Arithmetic programs-I/O**   In the case of arithmetic programs, we are mostly concerned with ensuring that they constitute a good model of "realistic" code. We observe a similar accuracy progression, across different models, as the one observed with MBPP and CRUXEval, and we can see that both input and output prediction become more difficult as the number of digits of the numbers fount in the program increases, as one would intutively expect. Finally, the benchmark is overall slightly more difficult than either MBPP and CRUXEval.

We now examine how code execution performance relates to the number of execution steps involved. Specifically, we analyze the success rate for varying numbers of steps across different LLMs. In Figure 4, we observe that performance degrades as the number of steps increases. Surprisingly, in some cases, the model fails even at problems with zero steps. The overall trend is that pre-trained models fail with increasing frequency at executing long programs.

**Discussion**   In section 2 we had remarked on the contrasting characteristics of the two benchmarks, chief among them the presence of *human intent* in the MBPP code, manifesting directly in the form of intuitive function names, and indirectly in the complexion of the code itself. In light of this, our results are not surprising: in order to predict the outputs and/or inputs of an MBPP code sample, the model has much less of a need to actually execute the code, and can instead take a shortcut by using the name of the function or just pattern-matching its code, thereby *guessing* the answer rather than *inferring* it. We provide further experimental support of this picture by running an additional set of evaluations, whereupon all function names are removed from the benchmark and the evaluation prompts. This ablation is referred to as "Anonymous Programs" in Figure 2; we can observe how performance (especially on output prediction) is negatively impacted by the absence of intuitive function names, although it usually remains above performance on the traced benchmark. An element of human intent is indeed still present in the function code itself, but this cannot be removed without fundamentally changing the nature of the benchmark.

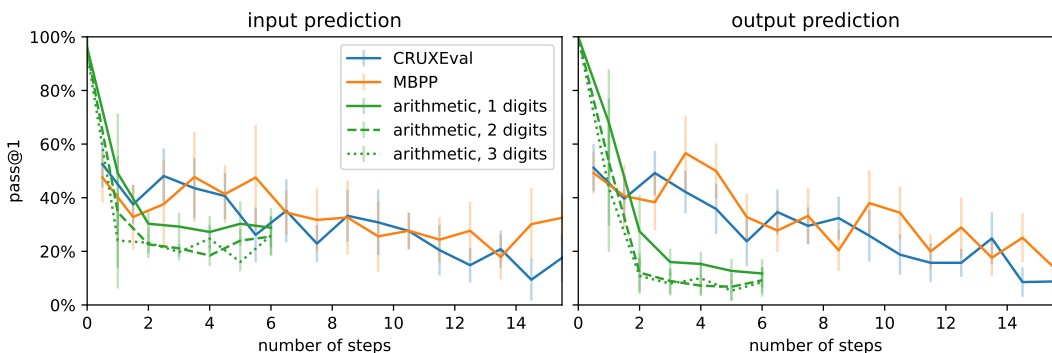

Figure 4: Performance per trace length: longer traces are harder to execute, presumably because LLMs have a harder time keeping track of state transitions. One aspect of difficulty/complexity.

Overall, these results are consistent with our belief that code traces constitute a more grounded benchmark of code execution, compared to the functions they come from.

## 4 STATE SUPERVISION

State tracking over multiple steps is a significant challenge in code execution. Our benchmark uncovers that longer traces, which involve multiple state transitions, result more frequently in execution failures. In this section, we explore how transformers can learn state tracking.

A Python interpreter updates the state after every execution step. This state information is however not part of the sequence of tokens the model is shown, as one can see in Table 2. A naive method to include this information is to interleave it with the code lines. We identify three issues with this approach. First, it requires modifying the prompt at inference time, which is challenging: one must decide when to insert new tokens for the state and when to continue reading the given prompt, and in most use cases it is undesirable to have to modify the prompt in this way. Second, this method forces the model to commit to a specific state, despite uncertainty in the prediction. Third, adding extra tokens increases the computational cost of training and inference, even for simple states. We propose an alternative method which incorporates state information without running into these issues.

We introduce an auxiliary training loss to optimize the predictability of the state *directly* from each line's hidden representations, while leaving the token sequence unaltered. While this slightly increases compute during training, it does not do so during inference. By retaining state information in the hidden representations rather than the token space, our method allows for expressing uncertainty over the state. This state supervision technique leverages state information without the drawbacks of adding it to the model's context.

Zaremba & Sutskever (2014) observe that learning code execution significantly benefits from a curriculum-based approach to generating training data. They first uniformly sample a hidden variable that controls the program's complexity and then generate a program based on this variable. For example, consider the parity problem: given a bit-string, predict whether the number of 1s is even or odd. Here, the length of the bit-string determines the problem's complexity. The curriculum first samples a length $n$ and then samples a bit string of that length. This method assumes control over the hidden variable that dictates complexity, which is often not possible for real-world programs. Nonetheless, we use this as our baseline, because we ourselves observe that without a curriculum, our transformer model struggles to learn effectively. Note that in our experiments, we apply the curriculum only in the baseline, and not in our state supervision method. In all the experiments in this section, we train a small transformer from scratch. See Appendix D for more details.

### 4.1 SINGLE TOKEN STATE

Our aim is to steer the hidden representations towards containing information about the state. To achieve this we take inspiration from linear probing (Alain, 2016; Belinkov, 2022)–a method that uses hidden representations during inference to train a linear classifier, checking whether these

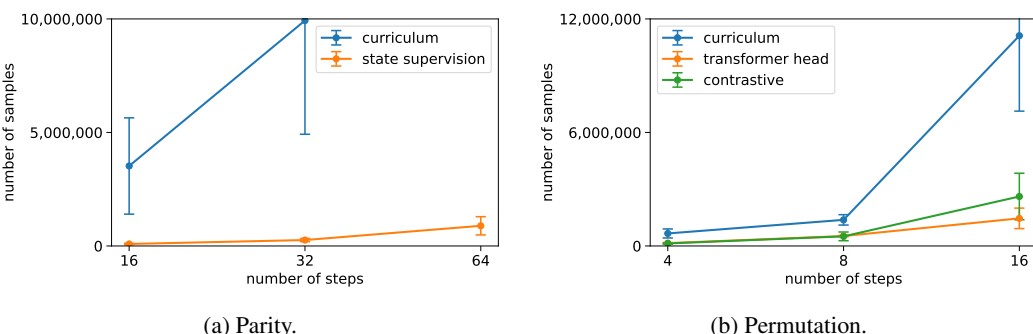

(a) Parity.  (b) Permutation.

Figure 5: Number of training samples to reach perfect test accuracy. Curriculum becomes infeasible for longer sequences, while state supervision scales favorably.

representations predict a feature of interest. Instead of adding this classification head only during inference, we incorporate it during training, and add its loss as an auxiliary loss to the standard next-token prediction objective.

Ideally, we want the model to process the code as it ingests the tokens–thinking while reading–rather than waiting until the question appears. To achieve this, we add state supervision to the last token of every line of code, which is typically the newline symbol.

Since state supervision only works to the degree to which single-line understanding is already present in the model, we initially restrict our attention to the simple Parity task, where the a single execution step is just a simple XOR operation between two single bits.

**Parity**  We formulate the Parity task as a code execution problem. See Figure 6 for an example. Here, the state is a single bit, and we add a linear classification head to predict the value of p after each line of code.

```
p = 1                 p = 1
p ^= 0                p = 1
p ^= 1                p = 0
assert p == ??
```

Figure 6: Parity example. On the left is the parity program, and on the right is the corresponding state at each line.

We evaluate the effectiveness of the state supervision method by comparing it against the curriculum baseline. The curriculum baseline trains on random bit strings of length 2 to $n$ while the state supervision method only trains on length $n$. Both evaluate on bit strings of length $n$ that are not seen during training. Since we can generate an arbitrary amount of training data, we expect any model to be able to achieve perfect accuracy. Hence, our metric shall be the number of seen samples needed reach perfect accuracy on the test set.

The results in Figure 5a highlight that solely using the next-token prediction objective is very inefficient for lengths 16 and 32, even with the curriculum. Furthermore, at length 64, the baseline did not improve upon the random baseline on the test set, despite training for $2^{25}$ (over 33 million) samples. Without the curriculum, the model would not even manage to learn parity for length 16. In contrast, state supervision solves parity for lengths 16 and 32 with only a fraction of the training samples. Additionally, we observe a significantly smaller increase in the number of samples required when increasing the length to 64. This experiment provides initial evidence for the effectiveness of state supervision. In the next section, we explore how to generalize this method to more complex states.

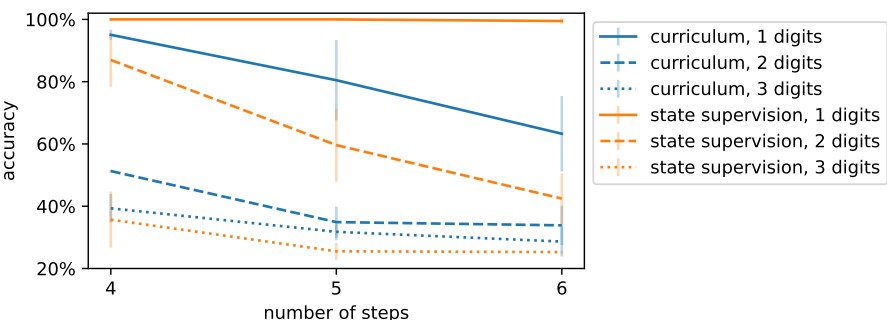

Figure 7: Test accuracy on Arithmetic Programs after training on $2^{23}$ samples. While still suffering from the two identified axes of difficulty, state supervision dominates curriculum.

## 4.2 Multi-token state

We consider a more generic case where the state consists of multiple tokens. To address this, we replace the linear classification head with one of two alternatives that can represent multi-token states. In one case, we use a small Transformer head to predict the state conditioned on the hidden representation. In the second, we apply a CLIP-like contrastive loss between the representation of the partial program, and that of its corresponding state. We compute the representation of states by applying the same language model on the state.

**Permutation**  We compare these two methods on *permutation programs*: the input is a list of numbers and the program applies a sequence of swaps between two elements in the list. Figure 8 shows an example program and the resulting state of each line.

```
a = [2, 1, 3]              a = [2, 1, 3]
a[0], a[1] = a[1], a[0]    a = [1, 2, 3]
a[1], a[2] = a[2], a[1]    a = [1, 3, 2]
assert a == ??
```

Figure 8: Permutation example. On the left is the permutation program, and on the right is the corresponding state of each line.

We set the list length to 4 and vary the number of swaps between 4 and 16. The higher the number of swaps, the higher the chance for an element to have swapped places multiple times. As a result, it becomes necessary to account for multiple code lines to predict the final location of a number. The Transformer head consists of a small 2-layer Transformer. For the contrastive auxiliary loss we need no additional parameters.

The results in Figure 5b demonstrate that state supervision remains effective for both the Transformer head and the contrastive method. This indicates that, despite not using a linear head, the auxiliary loss effectively induces a useful representation space in the language model. We observe that the contrastive setting requires lowering the factor on the auxiliary loss from 1 to 0.1. If the factor is too high, the loss decreases more slowly at the beginning of training because the representations are still uninformative, and the auxiliary loss adds a noisy signal.

**Arithmetic Program**  Finally, we test state supervision on Arithmetic Programs. From our results in section 3, we observed that pre-trained models already fail at programs of length 2. This indicates that pre-trained models struggle with these types of problems, despite likely having seen similar arithmetic problems in their training data.

We compare state supervision using a 2-layer Transformer as the state prediction head against the curriculum baseline. We evaluate different difficulty levels by varying the program length between 4 and 6 and the number of digits between 1 and 3. Increasing the number of digits increases the complexity of individual code lines. We expect state supervision to be effective in addressing the complexity added by longer programs, but not for more digits.

Since most models do not solve the test samples within a reasonable amount of training time, we set the number of training samples to $2^{23}$ and compare their final accuracy.

The results in Figure 7 show that state supervision effectively addresses the complexity added by increasing program length, while curriculum learning fails with increasing frequency for more number of steps. State supervision looses its efficacy in the 3 digit setting due to the difficulty of individual code lines. We expect that increasing the model size and number of training samples will recover the efficacy of state supervision.

**Discussion**   In all experiments in this section, we observe that the curriculum baseline underfits its most difficult training samples, which are the programs with the same number of steps as the test samples. Interestingly, even though the curriculum effectively adds out-of-distribution samples (programs with fewer steps), this results in an improvement on the in-distribution samples, despite reducing the number of in-distribution samples seen during training. However, the curriculum's effectiveness drastically decreases with longer programs, sometimes achieving no better than random performance.

Although state supervision adds an additional loss term to the language modeling objective, we observe that it improves the next-token prediction loss on the training set. This observation suggests that state supervision is not merely a regularization term. Instead, it highlights that executing state transitions step by step is an effective method for Transformers to execute code, and that state supervision induces this behavior in the Transformer.

## 5   RELATED WORK

**Code execution with LLMs**   Interest in the code generation capabilities of Language Models dates as far back as Language Models themselves, see e.g. the surveys (Zan et al., 2022) and (Fan et al., 2023). Benchmarks such as MBPP (Austin et al., 2021), HumanEval (Chen et al., 2021) and APPS (Hendrycks et al., 2021) have been designed to assess the capability of LMs to synthesize code from a natural language description. There are comparatively fewer benchmarks of code *execution* in LMs, though the problem itself is an old one, see e.g. (Zaremba & Sutskever, 2014). In (Gu et al., 2024), which was of inspiration for the present work, the authors introduce a benchmark of Python code execution, also uncovering some counter-intuitive insights on how code execution and code generation capabilities correlate to each other in transformer-based foundation models. We here take a step further on that path by introducing benchmarks of step-by-step execution of code; if an analogy between code execution and reasoning is made, our benchmarks can be seen as being meant to probe the extent to what a model *implicitly* executes code when prompted do to so (Wang et al., 2024), under the assumption that this allows for more reliability and generalizational capability compared to pattern-matching shortcuts.

**Interactive coding and task-solving**   Equipping an LLM with a Python Interpreter for the purpose of more reliably solving reasoning task has been proposed in (Chen et al., 2023; Gao et al., 2023), and more recently in (Li et al., 2023a). While our code traces are generated via such an interactive procedure, they are not meant as a way to let an LLM interact with a Python interpreter during inference, and they are not meant as an aid to carry out reasoning tasks; as a matter of fact, for two of our benchmarks (Traced CRUXEval and Arithmetic Programs) we make a point of generating traces from code which is lacking in task-solving intent. Another related work is (Zhang et al., 2024), where traces similar to our code traces are used to finetune an LLM to Behavioral-Clone standard Computer Science algorithms in an interactive form. The more general idea of coupling an LLM with an external module is also explored in e.g. (Ebrahimi et al., 2024); besides the module being a text scratchpad and not a Python interpreter, a qualitative difference lies in that the state of the scratchpad is directly ingested by the model together with its manipulation commands, whilst it remains implicit in our work.

**Execution traces and process supervision**   Related to the above is the use of execution traces as a form of CoT-like planning. In e.g. (Lehnert et al., 2024) and (Gandhi et al., 2024), the authors propose to train a model on execution traces of popular search algorithms in order to promote grounding and generalisation; in (Luo et al., 2024), the analogy with planning is taken further by supervising the execution trace via intermediate rewards. Our code traces are different from execution traces in

that they are executable and valid programs, whilst execution traces usually are not. Nevertheless, they can be seen as a form of CoT-like planning which decomposes a task (code execution) into its elementary steps, and we expressly designed them as a tool for promoting grounding.

**State tracking and auxiliary tasks in LLMs**   An examination of the state-tracking capabilities of modern LLMs and their relationship to code is given in (Kim et al., 2024), where it is observed that the capability of the model to track real-world entities (e.g. an apple) is improved when code is present in its training corpus. A major inspiration for the present work was (Li et al., 2023b), where is it found that the internal representations of a sequence model trained on the board game Othello are predictive on the current board state, to the degree that they enable causal intervention. A similar study (featuring real-world entities) was carried out in (Gurnee & Tegmark, 2023), where it was also found that the "period" token at the end of a sentence is the one carrying the most information on the state, similarly to the newline token in our code traces. The proposal of a training procedure meant to encourage the internalization of useful information in the model's hidden representation is the subject of refs. (Deng et al., 2023; 2024), though the focus is therein on reasoning steps rather than state information. Also of inspiration was (Gloeckle et al., 2024), wherein it is shown that training LLMs on a multi-token prediction task in addition to the usual next-token objective, improves their performance on the main next-token prediction task.

## 6   CONCLUSIONS AND LIMITATIONS

In this work, motivated by the observation that code is executed step-by-step in a classical computer, we introduce three new code execution benchmarks based on the unrolling of Python function codes into "code traces". Via these benchmarks, we observe that modern LLMs, when tested on code execution, sometimes take shortcuts rather than executing the code they are presented with, and that the main bottleneck they face when attempting code execution lies in limitations of their capability to track the state of the code. Motivated by this observation, we propose to enhance the training procedure of foundation models by adding an auxiliary state-prediction task on top of the usual next-token prediction objective, and empirically observe that this significantly boosts the learning efficiency of state-tracking behaviors.

Due to compute constraints, we could only evaluate our new benchmarks on small- to mid-size, open source LLMs; we are however confident that the trends and insights we observed will still be present when our benchmarks are evaluated on more capable and API-based LLMs, something that we strongly encourage readers to try. Due again to compute constraints, the experimental evaluation of our state-tracking enhancement procedure could only be carried out on a small (about 13M parameters) transformer model trained from scratch. Benchmarking our proposal during fine-tuning (e.g. via PEFT methods such as LoRA) of pre-trained LLMs is a task we leave for future work.

## REPRODUCIBILITY STATEMENT

We explain the general setup of our benchmark and experiments in the main paper and supplement necessary details for reproducibility in the appendix. Furthermore, we plan on releasing the code after acceptance.

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

## A  TRACED MBPP DATASET

The MBPP (Most Basic Python Programs) benchmark (Austin et al., 2021) is one of the oldest and most prevalent code generation benchmarks. It consist of simple python functions sourced from human programmers, each of them paired with a simple task description beginning with either "Write a function to..." or "Write a Python function to...". Along with each pair, three unit tests, generally in the form

```
assert <function name>(<input>) == <output>,
```

are provided in order to confirm that, given the task description, an LLM can generate a Python function solving the task. It is straightforward to turn MBPP into a code execution benchmark: for each function, we grab its inputs and outputs from the unit tests, meaning that in general (though not always, see below) one will end up with three benchmark samples per function. Afterwards, the inputs are used to run the function and thereby obtain their traced version. The details of the tracing procedure, which is the same as the one used on CRUXEval, are outlined in section 2. In order to minimize the chance of leakage, we carry out this procedure only on the test split of MBPP, consisting of 500 tasks and functions. Not of all these can be handled by our tracer, the reasons for this being:

- Multi-line conditional statements,

- Multi-line list, dictionary, or tuple comprehensions,

- No newline after a control flow element (e.g. `if:` or `else:`),

all of whom are not supported by out tracer. Moreover, we discard all functions whose execution runs longer that 50 execution steps, whose state traces run more than 12000 characters long, and whose code traces are longer than 1000 characters. These criteria also depend on the input the functions are supplied, and not just on the functions themselves. As a result, a function is not always run an all three inputs provided in the unit tests.

Eventually, we discard 173 functions of the 500 present, and we are left with 999 samples, containing 347 different functions.

## B  PROMPTS

### B.1  PROMPTS FOR PROGRAM PREDICTION ON CRUXEVAL

Our prompts for the program prediction task for CRUXEval are the same as those used in reference (Gu et al., 2024). Below is the one for the output prediction task:

```
[PYTHON]
def f(n):
    return n
assert f(17) == ??
[/PYTHON]
[ANSWER]
assert f(17) == 17
[/ANSWER]

[PYTHON]
def f(s):
    return s + "a"
assert f("x9j") == ??
[/PYTHON]
[ANSWER]
assert f("x9j") == "x9ja"
[/ANSWER]

[PYTHON]
<function_code>
assert f(<function_input>) == ??
[/PYTHON]
[ANSWER]
```

and here the one for the input prediction task:

```
[PYTHON]
def f(my_list):
    count = 0
    for i in my_list:
        if len(i) % 2 == 0:
            count += 1
    return count
assert f(??) == 3
[/PYTHON]
[ANSWER]
assert f(["mq", "px", "zy"]) == 3
[/ANSWER]

[PYTHON]
def f(s1, s2):
    return s1 + s2
assert f(??) == "banana"
[/PYTHON]
[ANSWER]
assert f("ba", "nana") == "banana"
[/ANSWER]

[PYTHON]
<function_code>
assert f(??) == <function_output>
[/PYTHON]
[ANSWER]
```

## B.2  PROMPTS FOR PROGRAM PREDICTION ON MBPP

In the MBPP case, we slightly tweak the CRUXEval prompts to include intuitive function names, in order to make them consistent with the presence of function names in the function codes. Here is the prompt we use for output prediction:

```
[PYTHON]
def identity(n):
    return n
assert identity(17) == ??
[/PYTHON]
[ANSWER]
assert identity(17) == 17
[/ANSWER]

[PYTHON]
def append_a(s):
    return s + "a"
assert append_a("x9j") == ??
[/PYTHON]
[ANSWER]
assert append_a("x9j") == "x9ja"
[/ANSWER]

[PYTHON]
<function_code>
assert <function_name>(<function_input>) == ??
[/PYTHON]
[ANSWER]
```

and below is the prompt for input prediction:

```
[PYTHON]
def count_even(my_list):
    count = 0
    for i in my_list:
        if len(i) % 2 == 0:
            count += 1
    return count
assert count_even(??) == 3
[/PYTHON]
[ANSWER]
assert count_even(["mq", "px", "zy"]) == 3
[/ANSWER]

[PYTHON]
def concat(s1, s2):
    return s1 + s2
assert concat(??) == "banana"
[/PYTHON]
[ANSWER]
assert concat("ba", "nana") == "banana"
[/ANSWER]

[PYTHON]
<function_code>
assert <function_name>(??) == <function_output>
[/PYTHON]
[ANSWER]
```

For the "unnamed" version of MBPP referred to in section 3, we just use the CRUXEval prompts reported in the previous section, while taking care to replace with "f" the function names contained in the function definitions.

### B.3 PROMPTS FOR THE TRACED BENCHMARKS

Our code traces do not contain any reference to function names. As a result, we can use the same prompts for traced CRUXEval, traced MBPP, and for our Arithmetic Programs, without the risk of inconsistencies. The one we employ for output prediction is simply a code-traced version of the CRUXEval program output prompt:

```
[PYTHON]
n = 17
assert n == ??
[/PYTHON]
[ANSWER]
17
[/ANSWER]

[PYTHON]
s = "x9j"
s += "b"
assert s + "a" == ??
[/PYTHON]
[ANSWER]
"x9jba"
[/ANSWER]

[PYTHON]
<function_trace>
[/PYTHON]
[ANSWER]
```

And the same logic is followed for the input prediction case:

```
[PYTHON]
n = ??
assert n == 17
[/PYTHON]
[ANSWER]
17
[/ANSWER]

[PYTHON]
s1, s2 = ??
s2 += "a"
assert s1 + s2 == "banana"
[/PYTHON]
[ANSWER]
"ban", "an"
[/ANSWER]

[PYTHON]
<function_trace>
[/PYTHON]
[ANSWER]
```

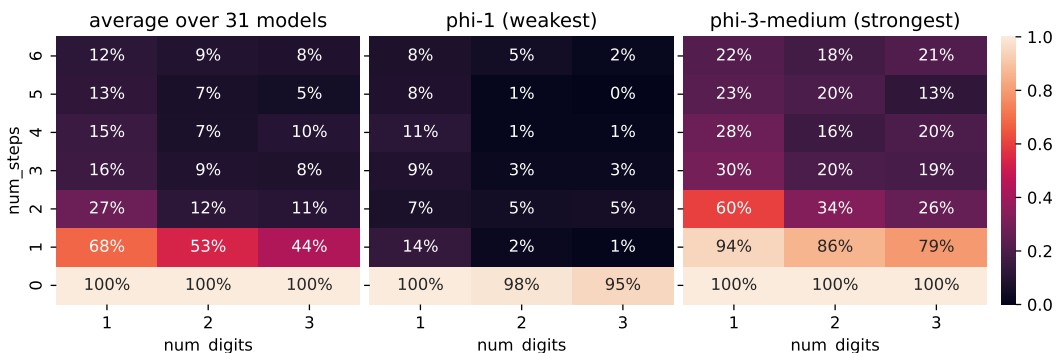

Figure 9: Effect of the two axes of difficulty.

## C  ARITHMETIC PROGRAM

We sample a random arithmetic program following these steps:

1. Set the output variable name to `a`

2. Sample a random expression
   - consisting of an operation (+ or -)
   - and two operands each of which is either a new variable, a free variable, or a random number
   - if a new variable is introduced, add it to the set of free variables

3. Randomly assign the expression to a free variable and close the variable if it does not appear in the expression

4. Repeat step 2 and 3 until the specified program length is reached

5. Close all free variables by assigning a random number to each

## D  STATE SUPERVISION EXPERIMENT DETAILS

All runs train a small Transformer with 6 layers and 512 hidden dimensions, resulting in 13641216 trainable parameters. The model follows the Llama architecture (Dubey et al., 2024). We employ the AdamW optimizer with a weight decay of 0.01. We use a cosine schedule for the learning rate, with a peak learning rate at 0.0003 and a warm up for 100 steps.

In the state supervision runs with the Transformer head, we apply the same architecture as we use for the language model, but with fewer parameters. It consists of 2 layers and 64 hidden dimensions. We condition this small Transformer on the hidden representation $h$ of the larger Transformer, by inserting $h$ as the initial token embedding. For the contrastive method, the objective is to retrieve the correct state's representation based on the representation of the partial program and vice-versa. We implement this in the form of a classification over states or partial programs. In our experiments we only consider states corresponding to the same program.

