# OpenReview forum: "Grounding code understanding in step-by-step execution"
_ICLR.cc/2025/Conference — Submitted to ICLR 2025_

### Official Review · Reviewer_yTY7 · 2024-10-22

**Soundness:** 2
**Presentation:** 2
**Contribution:** 3
**Rating:** 3
**Confidence:** 4

**Summary:**

In this paper, the authors propose a new approach to evaluate the code understanding capability of LLMs by predicting outputs (or inputs) from code traces. The authors convert two existing benchmark datasets, CRUXEval and MBPP, into code trace benchmarks to evaluate open-source LLMs with sizes ranging from 1B to 13B. In addition to these two existing benchmarks for code understanding and generation, the authors create a third benchmark focused on arithmetic programs to assess the LLMs from more perspectives. Moreover, besides evaluating the LLMs out-of-the-box, the authors introduce a novel auxiliary training loss to fine-tune decoder-only GPT-like models for reasoning over state transitions, which is demonstrated to be effective in predicting I/O from program traces.

**Strengths:**

* The issue that LLMs’ understanding of programs often relies on superficial aspects of the code, such as identifier names—which can be misleading and error-prone—is significant for both language understanding and software engineering. This paper addresses the problem by proposing a method to evaluate code understanding using traces.

* The idea of auxiliary state supervision loss is both novel and interesting.

**Weaknesses:**

* Presentation. Some sentences are hard to follow. For example, in the Introduction, "In other words, the code we consider only contains statements in each line, and as such, after each line the state of the program and its namespace are unambiguous."
* Novelty. The proposed code execution task to predict input or output ($(I, P) \mapsto O$ or $(O, P) \mapsto I$) seems  like a eased version of neural test generation [1,2], where the LLMs are asked to predict both input and output given a focal function/program ($P \mapsto (I, O)$).
* This paper assumes that LLMs understand and execute programs as a Turing machine. However, this assumption may or may not be valid, as LLMs might interpret code as other equivalent computational models. Given the current stage of research, we cannot firmly make this assumption. LLMs could also interpret programs as lambda calculus (Church) or general recursive functions (Godel).
* The authors claim that their transformation from source code to code trace is verifiable, but they do not provide details on the verification process. They state, *"We construct such step-by-step computations by unrolling the execution of a function on some input, while tracing the code lines."* However, it remains unclear how the authors ensure that these traces accurately represent the source code. In terms of program semantics, equivalence modulo inputs [3] suggests that different programs act the same for the same given input. Therefore, the soundness of this transformation approach cannot be confirmed without further explanation of the verification process.
* The authors filter out samples with traces longer than 50 steps. Why they only evaluate such simple programs? If the proposed step-by-step approach is valid, the variables at each step could be replaced or filled with the results from the previous step, making each step independent. Consequently, context length of the LLMs should not pose a problem. However, the authors still use the entire converted program as the prompt and observe performance degradation as the number of steps increases, which contradicts the intended concept of executing the program one step at a time.
* The approach of unrolling loops (as shown in Tables 1 and 2) assumes that LLMs understand the semantics of iterators and generators, which may not be true.
* It appears that the authors only expand loops but ignore syntactic sugar. For instance, in the arithmetic programs in Table 1, a statement like `d, a, c = 4, 3, 6` should be expanded into three separate lines: `d = 4; a = 3; c = 6` to maintain consistency with their earlier argument. The same principle should also be applied to the MBPP example in Table 2 and Figure 8.
* The auxiliary state supervision loss is insightful. However, the authors have not evaluated how this fine-tuning loss affects model performance on the original benchmark, i.e., without the transformation to code trace. Such an evaluation is crucial for assessing its effectiveness in real applications.
* The authors identify three issues with directly embedding state information in the input text to introduce the auxiliary loss. However, existing methods, such as the state monad transformer [4], allow stateful algorithms to be expressed in stateless programming language notations. Leveraging these techniques to incorporate state information into input embedding could potentially address the issues discussed in Section 4 while avoiding the shortcomings of fine-tuning the models with auxiliary loss.
* The auxiliary loss is given/defined clearly with necessary notation and equations. It is hard to understand how the loss is calculated at fine-tuning stage without proper equation.



[1]: Pengyu Nie, et al. *Learning Deep Semantics for Test Completion.* ICSE '23

[2]: Yifeng He, et al. *UniTSyn: A Large-Scale Dataset Capable of Enhancing the Prowess of Large Language Models for Program Testing.* ISSTA '24

[3]: Vu Le, et al. *Compiler Validation via Equivalence Modulo Inputs.* PLDI '14

[4]:  Shen Liang, et al. *Monad transformers and modular interpreters.* POPL '95

**Questions:**

Please address the concerns in **Weaknesses**.

---

### Official Review · Reviewer_qhTD · 2024-11-02

**Soundness:** 3
**Presentation:** 4
**Contribution:** 2
**Rating:** 5
**Confidence:** 4

**Summary:**

This paper aims to address the challenge that language models often struggle with accurately understanding code, as they tend to rely on shortcuts and memorization instead of following each step of code execution. The authors propose a new approach to evaluating code comprehension by assessing models abilities to predict state transitions line-by-line, using "code traces".

The authors enhance existing benchmarks by creating traced versions of CRUXEval and MBPP datasets, as well as a new synthetic Arithmetic Programs dataset. These traced benchmarks unroll each function into step-by-step instructions, capturing every intermediate state to provide a detailed view of code execution. Additionally, the authors propose a state supervision method that adds an auxiliary loss to improve the model’s ability to predict states at each step, leading to more accurate tracking of variable changes.

Experiments on these traced benchmarks show that code traces significantly enhance model accuracy, with improvements of up to 20-30% on CRUXEval input and output prediction tasks and consistent gains on Arithmetic Programs as task complexity increases. This demonstrates the potential of this approach to improve code understanding.

**Strengths:**

1) The paper is well organized and easy to follow
2) Paper introduces three new code execution benchmarks based on unrolling Python functions into code traces. By applying execution tracing to widely recognized benchmarks (e.g., CRUXEval, MBPP), the paper makes a valuable contribution by quantifying the effectiveness of state supervision.
3) The paper introduces an auxiliary loss on state prediction, which provides a novel framework for integrating state supervision at each execution step, leading to perf improvements.

**Weaknesses:**

It is important to note that the authors do not reference certain prior research on code execution with Transformer models and the use of execution traces, specifically:
(1) "Code Execution with Pre-trained Language Models", by C.Liu et. al (2023)
(2) "NExT: Teaching Large Language Models to Reason about Code Execution", by A. Ni et. al (April 2024)

Certain ideas in this paper build incrementally upon previous work in this area. In particular, CodeExecutor employed execution traces as a supervision signal in pre-training and considered curriculum learning for this task. Execution tracing served as a pre-training objective, enabling models to predict intermediate variable states and capture code's dynamic behavior. This approach to using execution traces for tracking variable states throughout program execution predates the "state supervision" method proposed in the current work.

**Questions:**

Comparison with CodeExecutor: Could the authors provide a detailed comparison of their model’s performance against CodeExecutor on similar benchmarks? Since CodeExecutor also utilizes execution traces as a supervision signal, even a qualitative comparison would help clarify the advantages of the proposed approach, particularly in terms of accuracy and state-tracking capabilities.

Could the authors elaborate on how their approach compares to NExT in terms of using execution traces for code understanding? Specifically, what benefits does the "state supervision" framework offer over NExT's CoT reasoning with execution traces, and how does it perform on similar program repair or execution reasoning tasks? General comments and avenues for improvement with step-by-step rationales would strengthen the related work discussion.

Swe-bench and LiveCodeBench are other examples of benchmarks that utilize code execution as a form of verification, and could be used to assess the benefit from inclusion of execution traces in-context of as a training task.

---

### Official Review · Reviewer_juDK · 2024-11-03

**Soundness:** 3
**Presentation:** 2
**Contribution:** 2
**Rating:** 3
**Confidence:** 4

**Summary:**

This paper studies the ability of LLMs on predicting the execution result of programs. It first evaluate 23 LLMs on CRUXEval, MBPP and arithmetic, letting the LLMs predicting either the input side or the output side. These LLMs produce relatively low Pass@k on the prediction. In details, on CRUXEval, LLMs predicts more accurate when faced with the traced version of a function than when faced with the original program. On MBPP, the trend is opposite that LLMs predict more accurate when faced with the original programs. And when the program becomes more complex with longer traces, the accuracy of LLMs' prediction on the execution also becomes lower.

To improve models' ability of understanding code execution, this paper proposes using explicit state supervision to induce step-by-step state transitions in the hidden representations. This approach is evaluated on single-token and multi-token states prediction. The state supervision approach outperforms the baseline curriculum approach on 1-digit and 2-digits states, yet underperforms on 3-digits states prediction.

**Strengths:**

1. This paper studies a interesting problem, a ability of predicting/understanding code execution result.
2. This paper evaluates a wide range of LLMs on CRUXEval, MBPP and Arithmetic programs, showing their ineffectiveness of predicting the execution states.

**Weaknesses:**

1. This paper is not well-organized, and somewhat fragmented in content. The first part of this paper (Section 2 and 3), the authors studies LLMs ability of predicting execution result, using CRUXEval, MBPP and Arithmetic programs. While in the second part (section 4), the proposed state supervision approach is only evaluated on Arithmetic programs using small Transformer models or classifiers. There is no clear relationships between these two parts, and how about applying the state supervision approach during the fine-tuning of LLMs? Can this approach be generalized to standard programs such as MBPP benchmark?
2. The effectiveness of state supervision approach is limited on 3-digits states prediction, which is unconvincing to prove the generalizability of the approach.

**Questions:**

See weaknesses.

1. Can the state supervision approach be applied during fine-tuning of LLMs?
2. Can this approach be generalized to standard programs such as MBPP benchmark?
3. The author needs more experiments to show the effectiveness and generalizability of the approach, as it only works on 1-digit and 2-digits states prediction, but not on 3-digits prediction.

---

### Official Review · Reviewer_AtF9 · 2024-11-03

**Soundness:** 2
**Presentation:** 1
**Contribution:** 2
**Rating:** 3
**Confidence:** 4

**Summary:**

The paper explores challenges in code understanding for large language models (LLMs), focusing on their limitations in tracking machine state during execution, which often leads to inaccuracies when predicting a program's behavior. To address this, the authors propose a step-by-step code execution benchmark that evaluates models based on their ability to trace line-by-line state transitions rather than relying on structural shortcuts. The study uses datasets derived from Python functions in the CRUXEval and MBPP benchmarks, with an additional synthetic set of Arithmetic Programs, to investigate LLM performance in both traced and untraced formats. Results show that LLMs tend to perform better on traced versions of CRUXEval but struggle with longer traces, revealing a semantic gap in their code understanding. To mitigate this, the authors introduce state supervision, an auxiliary training loss that encourages LLMs to incorporate state information within hidden representations. Empirical evaluations demonstrate that this approach improves state-tracking capabilities, though challenges remain, especially with more complex programs.

**Strengths:**

+ The paper proposes "code trace" as the intermediate representation between concrete program states and the source code, which seems to be new in the context of learning to understand code.

**Weaknesses:**

__Confusing Presentations__

In general, I feel the paper is difficult to understand in detail since most parts are described in a vague way, and the illustrated examples are missing. Also, the figures are sometimes incomplete: for example, the error bar of Figure-5.a is not fully included. I would strongly encourage the authors to polish the writing significantly and give examples to illustrate the methodology step by step for better understanding.

__Missing Important Baselines and Related Works__

While the authors claim the effectiveness of state supervision, they mostly compare it with its own variants and settings while ignoring those related existing works. For example, Scratchpad[1] already proposes a way to learn the concrete program states so that the LLM code learns to execute the program line by line. Most recently, NExT[2] proposed an improved version of scratchpad where they compact the concrete program states to learn the execution traces. Since the authors propose to unfold the execution, these formats of execution traces naturally become the baseline to compare with and illustrate the value of code traces proposed by this work. I would encourage the authors to perform fine-tuning experiments on these baselines as a comparison.

__Limited Insights On Top of Existing Works__

Existing works regarding the LLMs' limitation in understanding the code execution have been discussed for a while. They have evaluated the LLMs' capability from varied perspectives. For example, CRUXEval[3] reveals that LLMs are weak at predicting input/output, and instruction-tuned and chain-of-thoughts can hardly improve the accuracy; NExT[2] further illustrates LLMs can hardly understand program traces; this study [4] comprehensively studied such limitation from the perspectives for LLMs to predict (1) Code Coverage Prediction (2) Program States Prediction (3) Execution Path Prediction, and (4) Output Prediction. These related works have significant overlap with the conclusions drawn from this paper, so it is not clear what are the new insights drawn from this paper specifically. I would encourage authors to add a discussion section to discuss these existing studies and highlight their specific new insights for better understanding.

[1] Nye et al., Show Your Work: Scratchpads for Intermediate Computation with Language Models.

[2] Ni et al., NExT: Teaching Large Language Models to Reason about Code Execution.

[3] Gu et al., CRUXEval: A Benchmark for Code Reasoning, Understanding and Execution

[4] Chen et al., Reasoning Runtime Behavior of a Program with LLM: How Far Are We?

**Questions:**

- Can authors address the points raised in Weaknesses?

---

### Meta-Review · Area_Chair_wsNB · 2024-12-19

**Metareview:**

The paper studies LLMs' code understanding capability by evaluating their ability to predict execution states, focusing on line-by-line state transitions. It enhances existing benchmarks (CRUXEval, MBPP) by creating traced versions and introduces a synthetic Arithmetic Programs dataset. While the paper proposes an interesting state supervision approach for improving state-tracking capabilities, reviewers point out significant weaknesses in presentation clarity, missing baselines (e.g., Scratchpad, NExT), and limited new insights beyond existing works. The approach also shows limitations in handling complex states (3-digits) and lacks generalizability evidence.

**Additional Comments On Reviewer Discussion:**

The authors didn't provide a rebuttal.

---

### Decision · Program_Chairs · 2025-01-22

Reject